# Pandemic Experience of First Responders: Fear, Frustration, and Stress

**DOI:** 10.3390/ijerph19084693

**Published:** 2022-04-13

**Authors:** Ann Scheck McAlearney, Alice A. Gaughan, Sarah R. MacEwan, Megan E. Gregory, Laura J. Rush, Jaclyn Volney, Ashish R. Panchal

**Affiliations:** 1The Center for the Advancement of Team Science, Analytics, and Systems Thinking in Health Services and Implementation Science Research (CATALYST), College of Medicine, The Ohio State University, Columbus, OH 43210, USA; alice.gaughan@osumc.edu (A.A.G.); sarah.macewan@osumc.edu (S.R.M.); megan.gregory@osumc.edu (M.E.G.); laura.rush@osumc.edu (L.J.R.); jjvolney@gmail.com (J.V.); ashish.panchal@osumc.edu (A.R.P.); 2Department of Family and Community Medicine, College of Medicine, The Ohio State University, Columbus, OH 43210, USA; 3Department of Biomedical Informatics, College of Medicine, The Ohio State University, Columbus, OH 43210, USA; 4Division of General Internal Medicine, College of Medicine, The Ohio State University, Columbus, OH 43210, USA; 5Department of Emergency Medicine, The Ohio State University Wexner Medical Center, Columbus, OH 43210, USA

**Keywords:** COVID-19, first responders, emergency medical services, education, vaccination, mental health

## Abstract

Police officers, firefighters, and paramedics are on the front lines of crises and emergencies, placing them at high risk of COVID-19 infection. A deeper understanding of the challenges facing first responders during the COVID-19 pandemic is necessary to better support this important workforce. We conducted semi-structured interviews with 21 first responders during the COVID-19 pandemic, asking about the impact of COVID-19. Data collected from our study interviews revealed that, despite large numbers of COVID-19 infections among the staff of police and fire departments, some—but not all—first responders were concerned about COVID-19. A similar divide existed within this group regarding whether or not to receive a COVID-19 vaccination. Many first responders reported frustration over COVID-19 information because of inconsistencies across sources, misinformation on social media, and the impact of politics. In addition, first responders described increased stress due to the COVID-19 pandemic caused by factors such as the fear of COVID exposure during emergency responses, concerns about infecting family members, and frustration surrounding new work policies. Our findings provide insight into the impact of COVID-19 on first responders and highlight the importance of providing resources for education about COVID-19 risks and vaccination, as well as for addressing first responders’ mental health and well-being.

## 1. Introduction

First responders, including police officers, firefighters, and emergency medical services (EMS) personnel, are essential workers required to interact with the public, and in the midst of the Coronavirus disease 2019 (COVID-19) pandemic, their work has become particularly visible. Often at the forefront of the pandemic, these first responders have a high risk of exposure to COVID-19-positive individuals in the course of their job duties. Due to this increased risk of exposure, research has shown that first responders have a three-fold higher rate of COVID-19 infection compared to members of the general population [1].

Beyond their work-related exposure risk, the first-responder workforce has also faced other challenges due to the COVID-19 pandemic. Personal protective equipment shortages, particularly at the beginning of the pandemic, prevented some first responders from having appropriate equipment to protect themselves from infection [2,3]. In addition, with the high incidence of COVID-19 infection in first responders, there was the challenge of workforce depletion. For example, in the spring of 2020, nearly 17% of New York City’s police officers were unavailable due to illness or quarantine [4]. Similarly, New York City EMS and firefighters peaked at 19% and 13% of their workforces out on COVID-19-related medical leave, respectively [5]. Furthermore, the protests sparked by the deaths of Breonna Taylor, George Floyd, and others, which occurred predominantly in May and June of 2020 during the COVID-19 pandemic, also strained first responders as increased work hours were required and safety concerns around both violence and COVID-19 exposure were heightened [6].

Beyond the effects and consequences of COVID-19 infection, first responders have also experienced increased stress and anxiety, and research suggests that over half of frontline providers and first responders are concerned about their mental health in the context of the COVID-19 pandemic [7]. Specifically, first responders and frontline providers have reported increased feelings of sadness and anxiety, with a reluctance to ask for help [8]. First responders may also experience effects from being stigmatized due to their high exposure to COVID-19, leading to feelings of isolation [8]. Finally, social distancing, both on the job and in personal households, may affect first responders’ abilities to seek and receive support from their networks of coworkers, friends, or family [9]. Ultimately, the challenges faced by first responders during the COVID-19 pandemic may be contributing to a decreased likelihood that they remain in their occupation [10].

Notably, the rapid creation, increasing availability, and apparent effectiveness of vaccines against the SARS-CoV-2 virus [11], the virus that causes COVID-19, have begun to impact perspectives about COVID-19. However, vaccine hesitancy has made some individuals reluctant to receive a COVID-19 vaccine [12,13]. In addition, the recent emergence of the Omicron variant, which spreads more easily than both the original virus that causes COVID-19 and the Delta variant [14], has likely exacerbated first responders’ concerns in the midst of this pandemic. This variant resulted in a surge of cases between December 2021 and February 2022, with 99% of cases in the U.S. attributed to the Omicron variant [15]. Further, the Omicron variant has been reported to lead to increased numbers of breakthrough cases, wherein fully vaccinated individuals are more likely to acquire COVID-19 and spread it to others [15], compared to prior variants [14]. As such, first responders are at increased risk of becoming infected by the SARS-CoV-2 virus during their interactions with the public, ostensibly increasing anxiety and stress.

Given these myriad challenges affecting first responders and the continuing evolution of the COVID-19 pandemic, we conducted this study to learn how first responders feel about COVID-19 and its impact on their work. Our findings can help inform the development of educational resources about COVID-19 risks and vaccinations, as well as support efforts to address the challenges faced by first responders.

## 2. Materials and Methods

### 2.1. Study Setting and Population

We conducted interviews with first responders in a large midwestern city to learn about the impact of the COVID-19 pandemic on patrol officers, firefighters, and paramedics. We recruited 21 first responders who were employed at the Divisions of Police (PD) and Fire (FD) of one city and resided across 7 counties of the state.

Interviews were conducted from February to April 2021. During this time, COVID-19 tests were readily available, and vaccinations were underway for priority populations including health care personnel, first responders, residents and staff members of long-term care facilities, and the elderly.

In the study state, EMS personnel were eligible to receive the COVID-19 vaccine as a part of vaccination phase 1A, which began 14 December 2020. Law enforcement personnel with first-responder roles and individuals with an active firefighting certificate were eligible to receive the COVID-19 vaccine as a part of vaccination phase 1C, which began 4 March 2021 [16]. During vaccination phase 1A, individuals within the PD and FD could try to receive the vaccine at vaccination sites through a “do not waste” approach suggested by state officials. This approach permitted any vaccine unused due to missed vaccination appointments to be reallocated to other individuals.

### 2.2. Study Design, Data Collection, and Interview Procedures

With the help of liaisons in the PD and FD, approximately 2350 police employees and 1550 fire employees received information about our study through emails and flyers. Employees included department leaders, police officers, firefighters, paramedics, as well as other staff within the department. Interested individuals provided the study team with preferred times and methods of communication for an interview (i.e., phone or video). Interviewees were not compensated for participation. One-on-one interviews were then conducted using a semi-structured interview guide with open-ended questions asking about the following topics: (1) COVID-19 pandemic experience as first responders; (2) viral and serological testing; (3) vaccines; and (4) sources of information and guidance around COVID-19. On average, interviews lasted 30 min with a range of 18 to 61 min with interview length determined by the responses of the interviewee. The interviews were audio recorded and transcribed verbatim for analysis.

### 2.3. Data Analysis

Deductive-dominant thematic analysis was used to analyze all interview transcripts allowing for categorization of data based on general themes derived from the interview guides, as well as identification of emergent themes [17,18]. First, a preliminary coding dictionary was created based on the domains of the interview guide. Two study team members then coded 3 transcripts to confirm agreement, consistent application of the codes, and identification of any emergent codes. The finalized coding dictionary was used to code the remaining transcripts. This approach allowed for comparison of themes across all interviews and enabled us to characterize the impact of the COVID-19 pandemic on first responders. In addition, this constant comparative approach ensured that we reached saturation in our data collection with respect to the themes we present. ATLAS.ti software (ATLAS.ti Scientific Software Development, Berlin, Germany) was used to support the coding and data analysis process. To preserve anonymity, participant quotes were identified by a participant number.

## 3. Results

### 3.1. Characteristics of Study Participants

As noted in Table 1, the interviewees included 21 participants from PDs and FDs with a mean age (±SD) of 46 (±9) years. The majority were male (14/21; 67%) and employed by PDs (13/21; 62%). Interviewees were employed in a variety of roles, including department leader, firefighter, paramedic, and police officer. To protect the anonymity of participants, the classification of ‘Other’ was listed for specialized roles within PD and FD.

### 3.2. Impact of COVID-19 on Police and Fire Departments

During interviews, PD and FD leaders emphasized that COVID-19 had a major impact on first responders. A PD leader explained:*“The impact on our department’s been pretty huge. I mean we’ve had, I’m trying to think, overall, a third of our people get COVID. And, I think our positivity rate, our weekly positivity rate, is normally in the 30th percentile. Somewhere as high as 50% of the tests have been positive. So, we’ve had a lot of personnel impacted. Fortunately, we haven’t had any deaths, but we’ve had multiple hospitalizations of our employees.”**(Department leader, male, 50s.)*

Similarly, a FD leader shared:*“Huge impact. We have about 1600 firefighters. I think my numbers are right. We’ve had 485 positives in the department and we’ve had 1150 people that have been affected as far as they were contact traced or whatnot because of somebody at their station. So, let’s call that, I don’t know, 65% of our department… It’s been a great impact on everyone and just the change in their daily routine inside and outside the fire station.”**(department leader, male, 50s.)*

### 3.3. Perspectives about COVID-19 Infection

Despite the major impact of COVID-19 on their departments described by many PD and FD leaders, other interviewees had varied perspectives about COVID-19. Some first responders reported being very concerned about COVID-19. One police officer, for instance, expressed fear stating, *“It’s bizarre how it’s [COVID-19] going on and something that you know, we’ve never seen. I’ve never seen in my life before and it’s scary.” (Police officer, female, 50s.)* Others, including some department leaders, stated they were unconcerned and noted that they did not think of COVID-19 as a public health crisis. A department leader with this contrasting view reflected, *“I don’t know if I’m quite the right person to interview because I’m not worried about it one bit.*
*I’m not really buying into it. And I kind of do my thing and if I get it, I get it.” (Department leader, male, 40s.)* Additional representative quotations showing this variability in perspectives are listed in Table 2.

### 3.4. Perspectives about COVID-19 Vaccination

Similarly divided perspectives were found when interviewees were asked about COVID-19 vaccination. In this case, some interviewees clearly supported vaccination, such as a firefighter who shared, *“So, I think when you, when you look at the data, when you truly understand how the vaccine is made and what it does, I really, I had no issues, no hesitation of getting it.” (Paramedic, male, 50s.)* Others noted that they were not interested in getting the vaccine for reasons such as not trusting vaccine development and being afraid of side effects. A resistant police officer explained, *“We are not planning on getting one.*
*Well, I feel like it came out too fast. And there’s not enough known about it. I don’t get the flu shot either. I just don**’t feel comfortable getting it. I don**’t trust it. I**’m not letting my kid get it.**” (Police officer, female, 40s.)* One police officer similarly expressed apprehension about the vaccine’s development by reflecting, *“Mostly because it seemed like the process occurred pretty quickly.”* (Police officer, male, 30s.) Another resistant police officer explained, *“I had the whole COVID once and that was bad enough. And I’m like, the side effects I heard are pretty bad, but I just, I want more information before I get the vaccine.” (Police officer, female, 40s.)* Additional representative quotations are presented in Table 3.

### 3.5. Frustration around COVID-19 Information

When first responders were asked about where they get information on or hear about COVID-19, the great majority of our interviewees expressed frustration about three things: (1) COVID-19 information was reportedly confusing and inconsistent; (2) misinformation was pervasive on social media; and (3) politics appeared to have an impact on the information that was being shared. Examples of these areas of frustration are discussed next, with additional representative quotations presented in Table 4.

First, interviewees expressed frustration around inconsistencies in information that was being shared about COVID-19. One police officer explained that they felt that COVID-19 information was difficult to understand because, *“There’s so much information out there, you didn’t know what to believe. So, at that point, I literally had to quit watching the TV and quit reading stuff and just go with it.”*
*(Police officer, female, 50s.)* Similarly, another police officer expressed confusion stating, *“The inconsistency that you get from different entities, you know, you have Fauci saying one thing, the CDC saying another, and you know, doctors all over the place saying a myriad of different things.” (Police officer, male, 30s.)*

First responders also emphasized their frustration about COVID-19 misinformation found on social media. A police officer explained, *“It’s easy for somebody to put something on the Internet, somebody to read it, and share it, and it gets shared 10,000 times. If that’s your social group, that’s all you’re going to see. I think social media, it is what it is. It’s a demon.” (Police officer, male, 40s.)* A department leader similarly noted:*“You get these conspiracy theorists, like I’ll give you a conspiracy theory. Somebody out on social media. It happened pretty early on and it was shared by a bunch of people mostly people that are extreme right. …It was just way out there, just kind of really, really out there, conspiracy theory… I saw that one a couple different times from people and it was just like ‘What are you people believing?’”**(Department leader, male, 50s.)*

The impact of politics on COVID-19 information being shared was a third source of frustration reported by first responders. One police officer explained, *“I think that it’s hard, what you would deem genuine or real data or real information, because obviously it’s become politicized.” (Department leader, male, 50s.)* Similarly, a paramedic commented, *“So, I’ll be honest. I don’t give a damn what Dr. Fauci says. I don’t care what the Democrats say. I don’t care what the Republicans say. I just don’t give a [expletive] I am sick and tired of them all. It almost seems like every single one of them has got an agenda and I just don’t give a damn what it is.” (Paramedic, male, 40s.)*

### 3.6. Stress Brought on by COVID-19

Across interviews, a great majority of first responders acknowledged how the emergence of COVID-19 had increased their stress levels. Paramedics were particularly focused on the stress being faced by healthcare professionals. As one paramedic noted, *“I think the level of stress that healthcare workers, frontline, you know out in the community like me, as well as in the hospital, has gone up, oh man, hundreds of percentages. It’s taking its toll on a lot of people.” (Paramedic, male, 40s.)* Another paramedic reflected,*“Our run volumes increased, the number of patients at the hospital has increased, the number of mental health patients that we run on at the fire department and at the hospital have significantly increased. And those are always stressful runs, right? You’ve got someone losing their mind but at the same time you don’t want them to hurt you. And it’s just the level of stress has significantly gone up.”**(Paramedic, male, 40s.)*

Beyond the general increase in stress caused by performing their job duties during the COVID-19 pandemic, first responders across all roles also described three specific types of work-related stressors: (1) the risk of COVID-19 exposure during emergency responses; (2) concerns about passing a COVID-19 infection to family members; and (3) adapting to new work policies and procedures. More detailed examples of these sources of stress are discussed next, with additional representative quotations presented in Table 5.

First, comments made by interviewees underscored that many first responders were afraid of getting infected with COVID-19 while performing their job duties. One police officer noted, *“We can’t not go to something [an emergency call]. We have to go. So, it’s just an extra thing to put ourselves in the line of knowing we’re going out and dealing with people who potentially have COVID or are positive. It’s just an extra layer of stress on us.” (Police officer, male, 40s.)* Similarly, a paramedic explained:*“I have a job to do. Right? So, I just can’t stare at my patient and say I can’t help you because I’m afraid you’re going to give me COVID-19. That just, so I guess, I guess it’s kind of a risk-reward analysis that I take every time I’ve got a patient in the back of my squad. How big of a risk am I willing to take and what kind of a reward am I going to get? The reward is whether or not my patient gets better or gets worse.”**(Paramedic, male, 40s.)*

Fear of passing COVID-19 to family members was also highlighted as a stressor for first responders. One police officer reflected, *“I think the biggest thing is I’m just always worried about it. I was worried taking it home to my family members like some of the more high-risk people in my family or the baby at home.”*
*(Police officer, female, 30s.)* Similarly, a paramedic commented, *“Stress. You know, being worried about taking it home.” (Paramedic, male, 40s.)* Finally, a PD staff member reflected: *“I think there’s been stress too for the people who have to be at work all the time. Am I going to get this and what’s it going to do to me and my family?” (Other staff, female, 40s.).*

Third, many first responders emphasized that they experienced stress around adapting to new work policies and procedures during the pandemic. A paramedic explained:*“The difficulty lies in the way we approach runs now, is especially if it’s a run that has some fever issues or some respiratory issues, we go in without our partner the first whoever is completely dressed goes in and kind of talks to the person initially by yourself. So, I think things slow down a little bit, that makes it more difficult because you know, we’re problem solvers and we like to solve things fast in the fire department world. So, we walk up, and I start talking to you, my partner starts checking vital signs and you know, we have a flow that we have, and it’s definitely slowed the flow of our work down. So, you know, that’s certainly a difficulty that we experience.”**(Paramedic, male, 50s.)*

Similarly, a PD officer shared:*“We have to, you know, check our temperatures daily. There’s all the bull we have to go through daily, before we come to work. And then, when someone tests positive, then number one, there’s a whole trace, like going back on the trace and saying, ‘Were you in contact with them,’ and then that person has to get checked out for so long. Limiting the vacation times, so people, they’re getting kind of frustrated with that, and not being able to travel because if we go to a state where it’s over 15% [positivity rate] we are mandated to quarantine. And, we were having to use our own sick time for it. It’s just, you know, a lot of things going on with that aspect.”** (Police officer, female, 40s.)*

## 4. Discussion

The first responders we interviewed had varied perspectives about the COVID-19 pandemic, with some appearing to express little fear about the virus while others were quite concerned about both the possibility of becoming infected and exposing their family to the virus. Interestingly, first responder interviewees who did not perceive themselves to be personally at risk for COVID-19 spanned all age groups. This differs from the perspectives of the general public, where younger individuals perceive the risks of COVID-19 to be lower compared to older individuals [19]. Indeed, in the early stages of the pandemic, the elderly were at higher risk than younger people. Yet as no particular population or age group is without risk, and asymptomatic transmission remains an important factor in community spread, additional educational efforts to help people understand the virus and its transmission may be needed. This is likely especially important now as the highly transmissible Omicron variant is more likely to infect younger people than the original virus and other variants [14].

### 4.1. Barriers to COVID-19 Vaccination

Our study also identified several barriers to first responders’ willingness to get a COVID-19 vaccine: the personal belief that COVID-19 infection does not pose a significant threat to oneself and/or others, mistrust of the vaccine development process, and fear of vaccine side effects. Of note, the innovative messenger ribonucleic acid (mRNA) technology used to make the first two available vaccines is one factor that allowed the COVID-19 vaccines to be developed and deployed in clinical trials so quickly, but it has also been a subject of misunderstanding and a target of conspiracy theories on social media [20]. Moreover, at the time of our interviews the two most widely available vaccines had similar common short-term side effects (sore arm, fever, malaise) that were typically worse with the second dose [21,22], potentially contributing to interviewees’ reported concerns about short- and long-term side-effects as a reason to avoid COVID-19 vaccination.

### 4.2. Challenges to Understanding COVID-19 Information

The context of the COVID-19 pandemic has also contributed to challenges in understanding and adapting to evolving COVID-19 messaging, concerns which were noted by the first responders we interviewed. In addition to confusion about the messaging from information sources such as the federal government around COVID-19 [23], inconsistencies in messaging have been common as the science has evolved and recommendations have changed [24]. Widespread mis- and disinformation online has led to further confusion and skepticism [25,26], often resulting in additional stress [27]. The political nature of much of the COVID-19 discourse on social media and in the community has also contributed to this confusion and stress, echoed by results of a large survey demonstrating that political identity has a strong impact on one’s response to the COVID-19 pandemic [28].

### 4.3. Implications for First-Responder Workforce

The pandemic-related fear, frustration, and stress expressed by first responders in our study is aligned with previous work that identified occupational issues of concern (e.g., high rate of COVID-19 exposure, lack of PPE, breakdown of support systems) [1,2,3,9]. Our study also supports the need to address adverse effects on mental health, as previously noted by others [7,8,9]. It is possible that the continued stress of COVID-19 could contribute to a reduction in the first-responder workforce, as proposed by Hendrickson, et al. [10], and further exacerbate burnout, which was already a concern prior to the pandemic [29]. Given that the health and safety of the American public relies on a robust first-responder workforce, more research in this area is needed to understand resources and strategies that could be used to improve resilience and coping mechanisms in this group. In addition, professional organizations can be encouraged to provide leadership with consistent messaging and policy adaptations in response to updated public health information. Finally, fostering a work environment that encourages recognition of chronic stress may decrease stigma and strengthen camaraderie among first responders.

### 4.4. Limitations

Our study was limited to first responders working in one midwestern city; thus, the influence of any local, regional, and state health mandates and politics could not be considered. In addition, our study was not designed to explore potential differences related to role, age, or gender; thus, additional research to further understand the influences of these parameters on perceptions of first responders is warranted. Furthermore, this study did not include first responders who perform emergency services as volunteers in their communities. Additional research can be conducted to better inform our understanding of the perspectives of different types of first responders in the context of the COVID-19 pandemic.

## 5. Conclusions

Our study shows that, not surprisingly, the COVID-19 pandemic has significantly impacted first responders. First responders we interviewed expressed varied views about COVID-19 infection and vaccination, and described fear about ongoing risk of exposure, frustration about COVID-19 misinformation, and added stress due to the ongoing pandemic. These results highlight the importance of providing education on the safety and efficacy of COVID-19 vaccines (including mRNA technology), clarification and consistency around COVID-19 guidance and prevention strategies, and resources to address chronic stress and mental health issues for first responders working throughout the COVID-19 pandemic, as well as of continuing study to inform our understanding of how to best help first responders perform their critical roles in our emergency response system.

## Figures and Tables

**Table 1 ijerph-19-04693-t001:** Interviewee characteristics.

Participant Characteristics	Mean (±SD) or *n* (%)
Age (years)	
Mean (±SD)	46 (±9)
Range	29–56
Gender	
Male	14 (67)
Female	7 (33)
Division	
Police	13 (62)
Fire	8 (38)
Role	
Department leaders	4 (19)
Firefighters and paramedics	5 (24)
Police officers	8 (38)
Other	4 (19)

**Table 2 ijerph-19-04693-t002:** Perceptions about COVID-19 infection.

Perspectives about COVID-19 Infection	Comments from First Responders
Concerned	Another thing that worries me right now. I read the other day there is 4000 different strains, the variations of it. (Police officer, male, 50s.)
	I really try to not put myself in a situation where I’m going to have ongoing risk. (Other staff, female, 40s.)
Unconcerned	Like I wasn’t going to go actively breathe on people like when I had it, but I wasn’t afraid to get it. We just kind of rolled with it. We were just like, “If we get it, we get it.” (Police officer, female, 40s.)
	The more we learn about the virus, I’m, you know, relatively young and in good health. I don’t have any comorbidities. So, I think the more we learn about it and the more statistics that I saw, I think, the less concerned I got about it. Doesn’t mean I wasn’t still careful about reducing the spread, but I wasn’t necessarily afraid of the potential of exposure. (Police officer, male, 30s.)

**Table 3 ijerph-19-04693-t003:** Perceptions about COVID-19 vaccination.

Perspectives about COVID-19 Vaccines	Comments from First Responders
Supporter	I do, I talk to pretty much everybody at work, who I work around, or I see at work and encourage it [COVID-19 vaccine]. (Department leader, male, 50s.)
	We are all getting it, yeah. We were all on the no-waste list. And now that they just opened up to police today the remaining people in my section who never got called from the no-waste list are getting it. So, the whole section will have had it by the end of this week. (Other staff, female, 30s.)
Non-supporter	None of them [coworkers] are getting it. Um, I just don’t think they’re buying into the, you know, you need this rushed vaccine. (Department leader, male, 40s.)
	No, I have not gotten one and I don’t think I’m planning to. (Department leader, male, 50s.)
**Reasons for not getting vaccinated**	
Do not trust vaccine development	It just seemed like it was a little rushed to come out with a vaccine so quickly. (Police officer, female, 30s.)
	Let’s face it, they rushed this thing through really, really, fast. So, let’s kinda wait and let everybody else take it since I know I’m safe for now. Let’s let everybody else be the guinea pig. (Firefighter, male, 50s.)
Side effects	My biggest thing was just not knowing enough about what could potentially happen like in the future, if there were any long-term effects. That was like my biggest hold-up with it [COVID-19 vaccine]. (Firefighter, male, 30s.)
	Everybody was like, “Yeah, I got a sore shoulder for 24 h. Big deal.” Then everybody started getting the second shot and they look like shit for four days. And then everybody’s looking around at each other like, “Do I really want to feel like that?” (Paramedic, male, 40s.)

**Table 4 ijerph-19-04693-t004:** Confusion around COVID-19 information.

Perceptions about COVID-19 Information	Comments from First Responders
Confusing and inconsistent information	We need to be all consistent with what’s coming out. So, we’re all doing the same thing. People mix up something as simple as saying quarantine when they mean isolation and vice versa. So, there’s definitely a lot of misinformation and mis-education in my opinion. (Other staff, female, 40s.)
	The CDC guidelines is what we use. And I know for a first responder, that’s been a little bit challenging to understand sometimes. It’s changed from like as far as quarantine times, so that’s been somewhat confusing. (Police officer, male, 50s.)
Misinformation on social media	There’s not much you can do with the way social media is. I mean anybody can, you know, jump on any number of numerous platforms and write an article or a post or something and then that’s shared by millions of people. And, you know, it’s hard to tell what’s real and what’s not. I think in any walk of life, it’s going to be tough to combat the misinformation. (Police officer, male, 30s.)
	I think the greatest challenge in my personal world is sorting through the ridiculous beliefs that social media has put out as true. I feel very bad for people that believe a lot of the stuff that they quote-unquote “research” on social media. And I think people on social media are having a good time putting out ridiculous stuff just to see if people respond to it. And unfortunately, a lot of people believe that and buy into it. So, I think that’s really causing a lot of trouble with the ability to get the vaccine out and to push people in the direction of the vaccine. I feel like people are really getting a poor, poor representation of what the truth is through social media. (Paramedic, male, 50s.)
Impact of politics	It’s all like one big like political issue. You have people on the left and right who are, you know, they’ve fallen on certain sides of the whole debate. So, I mean, you know about vaccines. Like everything is, even like wearing masks, and it was, it all became political. (Firefighter, male, 30s.)
	I try to go to like CDC websites because I don’t trust any, either side of the news anymore, very much. I think both sides of the political spectrum slant the results to fit their agendas. So, I try to go with, with the CDC more than anybody else cause I figure, I mean, I’m sure there is some political crap there too, but I sort of trust them more than I trust anybody else. (Firefighter, male, 50s.)

**Table 5 ijerph-19-04693-t005:** Stress induced by COVID-19.

Work-Related Stressor	Comments from First Responders
Risk of COVID-19 exposure during emergency response	Quite a few members have gotten sick. Quite a few of them were, you know, pretty afraid to do their jobs and go out and do what they’re paid to do because they were worried about getting COVID. (Department leader, male, 40s.)
	It is affecting officers. The whole unknown of going into different houses and you don’t know if you’re having contact with someone positive. And now people, if they’re in trouble with the law, they want to say they’re positive, hoping that the officer will just go away. But we can’t in that aspect. (Police officer, female, 40s.)
Concerns about infecting family members	I worried about bringing it home to my son. Because they didn’t know at the time. That was my biggest concern. (Police officer, female, 50s.)
	My only concern has been giving it, bringing it home. And so, you know, I change my clothes and I shower at the fire station. I walk in the front door, I take those clothes off, and I get in the shower. (Firefighter, male, 40s.)
Adapting to new work policies and procedures	At the start I think some of my peers and the boss were not necessarily in belief that this is any different than the flu. And so for our staff there was like a split on enforcing or imposing a mask policy… the boss opted not to do it at that time. And I think in hindsight, though I thought at the time, and in hindsight, that was a mistake. It’s since got imposed, but I think that exposed a lot of people, plus set a tone that made it hard, harder to overcome when a policy did get imposed. (Department leader, male, 50s.)
	Losing time. Like we’ve gone through so many different iterations of leave policies and you know, there was a point where people had to use their own time and that’s stressful as well. (Other staff, female, 40s.)

## Data Availability

The data presented in this study are available on request from the corresponding author. The data are not publicly available due to participant privacy concerns.

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
