# Peer review of "Pandemic Experience of First Responders: Fear, Frustration, and Stress"

_ijerph, 2022, doi:10.3390/ijerph19084693_

Round 1
Reviewer 1 Report
Introduction
- There are no explicit relations to fear, stress, or frustration
- Stated that the perspectives of first responders are unknown however lines 50-79 reference studies that demonstrate that we do know what the perspectives of first responders are
Materials and methods
- It is unclear what questions were asked during the interviews
- They were not asked about fear, frustration, or stress explicitly
- The length of interviews likely did not allow enough time to fully explore the themes and achieve adequate results
- It is unclear what coding was used
Results
- It is unclear what constitutes “specialty staff”
- It is unclear why firefighters and paramedics were grouped together as they have different levels of risk and experiences that could contribute to different findings
- There is a large age range that is never addressed and likely contributes to important key differences which are not discussed
- The same is true of the gender difference
- Many of the themes presented are unrelated to fear, stress, and frustration
Discussion
- There is very little content relating to whether or not their results are consistent with current literature that was identified as existing in the introduction
- Increasing compliance to COVID-19 preventative measures is not related to the objectives of the study
- The limitations are lacking completeness
Conclusion
- There is no mention of fear
- There is no mention anywhere about why the findings highlight the importance of providing education and support to first-responders
Author Response
Reviewer #1 Comments and Suggestions for Authors
- Introduction
- There are no explicit relations to fear, stress, or frustration
Thank you for pointing out the relation of the study data to fear, stress, and frustration is unclear as written in our original paper submission. To illustrate the connection, we have now included the following language in our abstract:
“In addition, first responders described increased stress due to the COVID-19 pandemic caused by factors such as the fear of COVID exposure during emergency responses, concerns about infecting family members, and frustration surrounding new work policies.”
- Stated that the perspectives of first responders are unknown however lines 50-79 reference studies that demonstrate that we do know what the perspectives of first responders are
Thank you for pointing out that some of the perspectives of first responders are known and presented in the introduction of our paper. We have revised our abstract to provide better clarity about the goal of our study:
“A deeper understanding of the challenges facing first responders during the COVID-19 pandemic is necessary in order to better support this important work force.”
- Materials and methods
- It is unclear what questions were asked during the interviews
- They were not asked about fear, frustration, or stress explicitly
Thank you for pointing out that this information needs clarification in the Methods section. We have added the following text to the section including Study design, data collection, and interview procedures to note the interview topics more clearly:
“One-on-one interviews were then conducted using a semi-structured interview guide with open-ended questions asking about the following topics: 1) COVID-19 pandemic experience as first responders, 2) viral and serological testing, 3) vaccines, and 4) sources of information and guidance around COVID-19.”
Of note, questions did not explicitly ask about fear, frustration or stress but instead these themes emerged from participants’ responses. The persistence of these themes underlies the rationale for our emphasis on these topics.
- The length of interviews likely did not allow enough time to fully explore the themes and achieve adequate results
Our study team consists of highly experienced interviewers who allowed interviewees to share their perspectives on each of the interview topics. While we respectfully disagree that these interviews were not long enough, we have added the following text to the Methods section to clarify the length of interviews:
“On average, interviews lasted 30 minutes with a range of 18 to 61 minutes with interview length determined by the responses of the interviewee.”
- It is unclear what coding was used
Thank you for pointing out that our coding method needs clarification in the Methods section. We have added the following text to the Data analysis section to describe the coding method and analysis more clearly:
“Deductive-dominant thematic analysis was used to analyze all interview transcripts allowing for categorization of data based on general themes derived from the interview guides, as well as identification of emergent themes… This approach allowed for comparison of themes across interviews and enabled us to characterize the impact of the COVID-19 pandemic on first responders. In addition, this constant comparative approach ensured that we reached saturation in our data collection with respect to the themes we present.”
- Results
- It is unclear what constitutes “specialty staff”
Thank you for pointing out that the term “specialty staff” needs clarification in the Results section. We have added the following text to the Results section to describe the interviewees more clearly:
“To protect the anonymity of participants, the classification of “Other” was listed for specialized roles within PD and FD.”
- It is unclear why firefighters and paramedics were grouped together as they have different levels of risk and experiences that could contribute to different findings
- There is a large age range that is never addressed and likely contributes to important key differences which are not discussed
- The same is true of the gender difference
Thank you for this feedback regarding the potential differing perspectives related to role, age, and gender. We agree that differing perspectives related to role, age, and gender may exist, and note that associated with each quote presented is the interviewee’s role and gender. Our study was not designed to examine potential differences between roles, age, and gender but instead the topics we describe in this paper. Such differences may be investigated as part of an expanded study.
- Many of the themes presented are unrelated to fear, stress, and frustration
As noted above, interview questions did not explicitly ask about fear, frustration or stress but instead these themes emerged from participants’ responses. The persistence of these themes underlies the rationale for our emphasis on these topics.
- Discussion
- There is very little content relating to whether or not their results are consistent with current literature that was identified as existing in the introduction
Thank you for this feedback. We have included a paragraph in the discussion that relates our findings to current literature cited in the introduction.
“The pandemic-related fear, frustration, and stress expressed by first responders in our study is aligned with previous work that identified occupational issues of concern (e.g., high rate of COVID-19 exposure, lack of PPE, breakdown of support systems). Our study also supports the need to address adverse effects on mental health, as previously noted by others. It is possible that the continued stress of COVID-19 will contribute to a reduction in the first responder workforce, as proposed by Hendrickson, et al., and further exacerbate burnout, which was already a concern prior to the pandemic. Given that the health and safety of the American public relies on a robust first responder workforce, more research in this area is needed to understand resources and strategies that could be used to improve resiliency and coping mechanisms in this group.”
- Increasing compliance to COVID-19 preventative measures is not related to the objectives of the study
Thank you for this feedback. Upon further reflection, we agree that this section on compliance is not related to the study objectives, and we have removed it from the revised document.
- The limitations are lacking completeness
Thank you for this feedback. We have revised the limitations section to include information about the potential influences of role, age, and gender on our results. We also note that local, regional, and state health mandates could limit generalizability.
“Our study was not designed to explore potential differences related to role, age, or gender, thus additional research to further understand the influence of these parameters on perceptions of first responders is warranted.”
- Conclusion
- There is no mention of fear
Thank you for this feedback, we have revised the conclusion to provide more information about our findings specific to the topic of fear:
“First responders we interviewed expressed varied views about COVID-19 infection and vaccination, and described fear about ongoing risk of exposure, frustration about COVID-19 misinformation, and added stress due to the ongoing pandemic.”
- There is no mention anywhere about why the findings highlight the importance of providing education and support to first responders
Thank you for this feedback. We have clarified the section on providing education and support as follows:
“…providing education on the safety and efficacy of COVID-19 vaccines (including mRNA technology), clarification and consistency around COVID-19 guidance and prevention strategies, and resources to address chronic stress and mental health issues for first responders working throughout the COVID-19 pandemic…”
Reviewer 2 Report
Dear Authors,
Thank you for this novel study on the impact of COVID-19 on first responders. It is unique and can make an important contribution to the literature related to this specific population. I believe your manuscript is quite good, and have a few suggestions and clarifications for your consideration:
- Consider replacing instances of "currently" (line 77) or "especially important now" (line 277) with a specific date/timeframe to make is easy for a reader to know the point of time reference when reading this article months, or years, from now.
- Clarify police and fire employees definition in your study design: At line 107/108 you mention recruitment emails sent to police and fire employees, which may make a reader wonder about the paramedics. You talk about first responders, and police, fire, and paramedics up until this point, and someone (not from the U.S.) might wonder about the paramedics and not know that they are frequently employed by the fire department.
- In your data anlaysis (line 119/120) you mention emergent codes. It appears that you only identified emergent codes in 3 transcripts as you constructed your coding dictionary and did not look for them beyond that? Can you clarify?
- Interviewee characteristics: Table 1 breaks out the role of your interviewees, but nowhere is it clearly defined what consititutes a department leader or speciality staff. Similar to what I said above about fire/police/paramedics, can you consistently identify your sample in this way? How many department leaders were FD vs. PD, same with specialty staff.
- Section 3.6 seemed to be particularly quote-heavy and I wonder if you could reduce the quotes slightly?
- Not to contradict the quote-heavy comment I made above, but you do discuss the stress brought by COVID-19 in the results, but make no mention of it in your discussion. Given the literature and emerging data about burnout, might you work stress into the discussion?
I wish you all the best with this manuscript.
Author Response
Reviewer #2 Comments and Suggestions for Authors
Thank you for this novel study on the impact of COVID-19 on first responders. It is unique and can make an important contribution to the literature related to this specific population. I believe your manuscript is quite good.
Thank you for this positive feedback.
I have a few suggestions and clarifications for your consideration:
- Consider replacing instances of "currently" (line 77) or "especially important now" (line 277) with a specific date/timeframe to make is easy for a reader to know the point of time reference when reading this article months, or years, from now.
Thank you for this suggestion, we have added the following text to give context to the point of time reference:
“This variant resulted in a surge of cases between December 2021 and February 2022, with 99% of cases in the U.S. attributed to the Omicron variant.”
- Clarify police and fire employee definition in your study design: At line 107/108 you mention recruitment emails sent to police and fire employees, which may make a reader wonder about the paramedics. You talk about first responders, and police, fire, and paramedics up until this point, and someone (not from the U.S.) might wonder about the paramedics and not know that they are frequently employed by the fire department.
Thank you for this suggestion, we have added the following text to clarify the recipients of our study recruitment email:
“With the help of liaisons in the PD and FD, approximately 2,350 police employees and 1,550 fire employees received information about our study through emails and flyers. Employees included department leaders, police officers, firefighters, paramedics, as well as other staff within the department.”
- In your data analysis (line 119/120) you mention emergent codes. It appears that you only identified emergent codes in 3 transcripts as you constructed your coding dictionary and did not look for them beyond that? Can you clarify?
As explained above, we have added the following text to the Data analysis section to describe the coding method and analysis more clearly:
“Deductive-dominant thematic analysis was used to analyze all interview transcripts allowing for categorization of data based on general themes derived from the interview guides, as well as identification of emergent themes… This approach allowed for comparison of themes across all interviews and enabled us to characterize the impact of the COVID-19 pandemic on first responders. In addition, this constant comparative approach ensured that we reached saturation in our data collection with respect to the themes we present.”
- Interviewee characteristics: Table 1 breaks out the role of your interviewees, but nowhere is it clearly defined what constitutes a department leader or specialty staff. Similar to what I said above about fire/police/paramedics, can you consistently identify your sample in this way? How many department leaders were FD vs. PD, same with specialty staff.
Thank you for this feedback regarding our interviewee characteristics. As described above, we wanted to protect the anonymity of participants. Given our sample size of 21 interviewees, a further breakdown of their roles could risk their identification. We have clarified this further as follows:
“To protect the anonymity of participants, the classification of “Other” was listed for specialized roles within PD and FD.”
- Section 3.6 seemed to be particularly quote-heavy and I wonder if you could reduce the quotes slightly?
We appreciate your suggestion but respectfully decline to reduce the quotes in this section.
- Not to contradict the quote-heavy comment I made above, but you do discuss the stress brought by COVID-19 in the results, but make no mention of it in your discussion. Given the literature and emerging data about burnout, might you work stress into the discussion?
Thank you for this feedback. As noted in response to Reviewer #1, we have added a section in our revised discussion section that addresses stress and burnout more completely.
“The pandemic-related fear, frustration, and stress expressed by first responders in our study is aligned with previous work that identified occupational issues of concern (e.g., high rate of COVID-19 exposure, lack of PPE, breakdown of support systems). Our study also supports the need to address adverse effects on mental health, as previously noted by others. It is possible that the continued stress of COVID-19 could contribute to a reduction in the first responder workforce, as proposed by Hendrickson, et al., and further exacerbate burnout, which was already a concern prior to the pandemic. Given that the health and safety of the American public relies on a robust first responder workforce, more research in this area is needed.”
Reviewer 3 Report
Overview:
The authors conducted a qualitative study to examine the perspectives of first responders regarding the COVID-19 pandemic, COVID-19 testing procedures, and vaccines. The authors offer compelling evidence to suggest first responders are divided in their perspectives on these topics, and that they experience multiple stressors related to COVID-19. I provide minor comments below with the aim of helping the authors prepare a revision.
Specific Comments:
Methods
2.1 Study setting and population
- I believe the authors can provide a sentence or two to clarify their sampling approach. Did they purposefully sample first responders and department leaders from a single city in the state? Did they approach other departments in other cities or counties, but those departments declined to participate or did not respond to requests to participate?
2.2 Study design, data collection, and interview procedures
- Were participants incentivized to participate in the interviews or were they compensated for participating in this study? If yes, how were they compensated? If no, please explicitly state that they were not compensated.
Results
3.1 Characteristics of study participants
- Paragraph starting, “As noted in Table 1…”: What is an example of a “specialty staff” member?
Discussion
4.4 Limitations
- First sentence: Change “first responders from one midwestern city” to “first responders working in one midwestern city” to clarify that first responders were not identified nor recruited based on their city or county residence.
- I think the authors can add a few sentences about the future research implications of their study. For example, since this study focused on individuals’ perspectives, I believe this study highlights an opportunity for future research to examine the workplace’s role in disseminating COVID-19-related information. How can department leaders address the concerns of staff to mitigate the negative impact of this crisis on staff morale? What do we need to know more about to better prepare departments for future crises?
Author Response
Reviewer #3 Comments and Suggestions for Authors
The authors conducted a qualitative study to examine the perspectives of first responders regarding the COVID-19 pandemic, COVID-19 testing procedures, and vaccines. The authors offer compelling evidence to suggest first responders are divided in their perspectives on these topics, and that they experience multiple stressors related to COVID-19.
Thank you for this positive feedback.
I provide minor comments below with the aim of helping the authors prepare a revision:
- Methods
2.1 Study setting and population
- I believe the authors can provide a sentence or two to clarify their sampling approach. Did they purposefully sample first responders and department leaders from a single city in the state? Did they approach other departments in other cities or counties, but those departments declined to participate or did not respond to requests to participate?
Thank you for this feedback. We have added the following text to describe study setting and population more clearly:
“We recruited 21 first responders who were employed at the Divisions of Police (PD) and Fire (FD) of one city and resided across 7 counties of the state.”
2.2 Study design, data collection, and interview procedures
- Were participants incentivized to participate in the interviews or were they compensated for participating in this study? If yes, how were they compensated? If no, please explicitly state that they were not compensated.
Thank you for this suggestion, we have added the following sentence to section 2.2:
“Interviewees were not compensated for participation.”
- Results
3.1 Characteristics of study participants
- Paragraph starting, “As noted in Table 1…”: What is an example of a “specialty staff” member?
Thank you for pointing out that the term “specialty staff” needs clarification in the Results section. As described above, we have added the following text to the Results section to describe the interviewees more clearly:
“To protect the anonymity of participants, the classification of “Other” was listed for specialized roles within PD and FD.”
- Discussion
4.4 Limitations
- First sentence: Change “first responders from one midwestern city” to “first responders working in one midwestern city” to clarify that first responders were not identified nor recruited based on their city or county residence.
Thank you for this suggestion, we have revised the first sentence accordingly.
- I think the authors can add a few sentences about the future research implications of their study. For example, since this study focused on individuals’ perspectives, I believe this study highlights an opportunity for future research to examine the workplace’s role in disseminating COVID-19-related information. How can department leaders address the concerns of staff to mitigate the negative impact of this crisis on staff morale? What do we need to know more about to better prepare departments for future crises?
Thank you for this feedback. We have added the following section that addresses future research and leadership implications.
“Given that the health and safety of the American public relies on a robust first responder workforce, more research in this area is needed to understand resources and strategies that could be used to improve resilience and coping mechanisms in this group. In addition, professional organizations can be encouraged to provide leadership with consistent messaging and policy adaptations in response to updated public health information. Finally, fostering a work environment that encourages recognition of chronic stress may decrease stigma and strengthen camaraderie among first responders.”
Reviewer 4 Report
It is an interesting study regarding the perception of some people from three fields that are `in the first line` in the confrontation with covid 19 and its derivatives.
However, some aspects I think must be clarified:
- Two pieces of information are a bit confusing and contradictory:
- At page 3, point 2.2, lines 108-109, you said „Interested individuals provided the study team with preferred times and method of communication for an interview”
- At page 9, point 4.4 lines 316-317, you said: „In addition, this study did not include volunteer first responders”
In my opinion, answering whoever wants means actually volunteering.
- It should still be specified what was the number of people to whom the initial email was sent by those „key contacts”: page 3, rows 107-108: „recruitment emails were sent to police and fire employees”. Exactly how many emails were sent? What was the response rate?
- How those “key contacts” had access to the respondents' email addresses and who gave them the right to use them? This should be clarified. And maybe the expression „key contacts” it is not the most appropriate one - it's like we're in an action movie, with suspects and things like that J
Author Response
Reviewer #4 Comments and Suggestions for Authors
It is an interesting study regarding the perception of some people from three fields that are `in the first line` in the confrontation with covid 19 and its derivatives.
Thank you for this positive feedback.
However, some aspects I think must be clarified:
- Two pieces of information are a bit confusing and contradictory:
- At page 3, point 2.2, lines 108-109, you said „Interested individuals provided the study team with preferred times and method of communication for an interview”
- At page 9, point 4.4 lines 316-317, you said: „In addition, this study did not include volunteerfirst responders” In my opinion, answering whoever wants means actually volunteering.
Thank you for this feedback, we have revised section 4.4 to provide more information about the first responders who participated in our study:
“In addition, this study did not include first responders who perform emergency services as volunteers in their communities. Additional research can be conducted to better inform our understanding of the perspectives of different types of first responders in the context of the COVID-19 pandemic.”
- It should still be specified what was the number of people to whom the initial email was sent by those „key contacts”: page 3, rows 107-108: „recruitment emails were sent to police and fire employees”. Exactly how many emails were sent? What was the response rate?
Thank you for this feedback. We have revised Section 2.2 to include details about how many employees received information about our study:
“With the help of liaisons in the PD and FD, approximately 2,350 police employees and 1,550 fire employees received information about our study through emails and flyers. Employees included department leaders, police officers, firefighters, paramedics, as well as other staff within the department.”
- How those “key contacts” had access to the respondents' email addresses and who gave them the right to use them? This should be clarified. And maybe the expression “key contacts” it is not the most appropriate one - it's like we're in an action movie, with suspects and things like that J
Thank you for this feedback. As described above, we have revised the methods section to clarify that the key contacts were liaisons within the police and fire departments.